This is a Registered Report and may have an associated publication; please check the article page on the journal site for any related articles.

**Data Availability Statement:** All relevant data from this study will be made available upon study completion.

REGISTERED REPORT PROTOCOL

# Antimicrobial efficacy of silver nanoparticles against Candida albicans: A systematic review protocol

**Razia Z. Adam, Saadika B. Khan** *

Department of Restorative Dentistry, Faculty of Dentistry, University of the Western Cape, Cape Town, South Africa

* skhan@uwc.ac.za

## Abstract

### Introduction

Denture-induced stomatitis is one form of candidiasis. It is characterised as inflammation and erythema of the oral mucosa underneath the denture-bearing areas and clinically classified into three types according to severity. Denture hygiene, appropriate mouth rinses and the use of antifungal therapy are commonly used to treat the condition, but new technologies are emerging that may assist in its treatment.

### Aim

The aim of this systematic review is to determine if silver nanoparticles inhibit the growth of *Candida Albicans* when included in acrylic dentures and in different denture liners.

### Methodology

A protocol was developed and published on PROSPERO (Registration No: CRD42019145542) and with the institutional ethics committee (Registration No: BM20/4/1).

The protocol includes all aspects of a systematic review namely: selection criteria, search strategy, selection methods using predetermined eligibility criteria, data collection, data extraction, critical appraisal of included studies, and the intended statistical analyses such as calculating risk ratios (RR) for dichotomous outcomes and presented at 95% confidence intervals, a meta-analysis, if possible or a narrative report as needed.

### Expected results

With rigorous inclusion criteria set and databases identified for searching, appropriate clinical and laboratory studies may be obtained but the results and its interpretation and translation into clinical practice may be a challenge as these depend on the quality of the research.

**Funding:** The authors received no specific funding for this work.

**Competing interests:** The authors have declared that no competing interests exist.

# Background

Oral diseases now rank in the top ten leading causes of years lived with disability (YLDs) [1]. The United Nations released the key findings of the World Population Prospects in 2019 and they estimate that life expectancy will increase to 77 years [2]. In addition, by 2030 older persons are expected to outnumber children under the age of 10 years. This increase in the elderly population means that more individuals may lose their teeth. In 2013, it was reported that edentulism accounted for a third of the burden of disease in low and middle income countries globally [3]. Although the prevalence of edentulism is decreasing in high-income countries, the opposite is experienced in low-income countries. More importantly, non-communicable diseases and edentulism share the same risk factors such as smoking, alcohol use and unhealthy dietary behaviours [3]. The lack of access or limited access to dental care is strongly associated with edentulism [3]. In low-income countries, the early loss of teeth due to extractions are common. Extraction of the offending tooth is often the only acceptable, affordable and available option to treat carious teeth for these communities. The resulting loss of teeth also influences speech, aesthetics, the ability to eat and socialise.

In South Africa, Peltzer et al. (2014) reported that 3–9% of adults over the age of 50 years were completely edentulous [4]. The rehabilitation of edentulous patients involves the construction of a removable prosthesis. The placement of a denture into the oral cavity introduces a surface for the development and adhesion of a dental biofilm. The imbalance of oral microbes in the oral cavity is then further affected by the advanced age of the patient as well as any pre-existing systemic conditions [5]. *Candida* is a normal oral commensal and found in 35–80% of healthy mouths [5]. However, any imbalance may cause a proliferation of *Candida* resulting in a pathogenic relationship.

## Description of the condition

Denture-induced stomatitis is one form of candidiasis. It is characterised as inflammation and erythema of the oral mucosal areas covered by the denture [6]. *Candida albicans* is the most common causative pathogen of oral candidiasis, the disease associated with a fungal infection [6]. Oral candidiasis may also occur in patients with predisposing factors, such as an immuno-compromised status, xerostomia, age of the denture and poor oral hygiene [7]. Denture hygiene, appropriate mouth rinses and the use of antifungal therapy are commonly used to treat the condition [7].

## Description of the intervention

Nanotechnology has emerged as a promising technique for various biomedical applications. Metal oxide nanoparticles well known for their antibacterial effect include silver (Ag), iron oxide ($Fe_3O_4$) titanium oxide ($TiO_2$), copper oxide (CuO) and zinc oxide (ZnO) [8]. Silver nanoparticles (AgNPs) destroy all pathogenic microorganisms and no resistance to its actions has been reported [7]. Silver nanoparticles have also been found to be non-toxic to humans, and very effective against bacteria and viruses at very low concentrations with no side effects [9].

The preparation of metal nanoparticles (NPs) involves a variety of chemical and physical methods such as chemical-, photochemical- and electrochemical-reduction, laser ablation and lithography [10]. The use of these chemicals may limit their use in biomedical applications [10].

The green synthesis of silver nanoparticles is readily available, is safer and contain a variety of reducing phytochemicals [10]. Biological methods are performed in eco-friendly conditions and consume no energy [11]. The biological methods are cost-effective, environmentally

friendly and include a single step process for large scale synthesis of nanoparticles [11]. With the biological method there is no need to use high pressure, energy, temperature and toxic chemicals which are harmful to the health of living organisms [11]. The synthesis of silver nanoparticles by using plant extracts are better than other biological methods [11]. The intricate process of maintaining cell culture can be eliminated as silver nanoparticles produced by plant extracts are usually more stable and more varied in shape and size in comparison with those produced by other methods [11].

Silver nanoparticles have been used in dentistry in a number of ways and within different systems: composite resins and adhesive systems, acrylic resin, endodontic materials and titanium implants. Biosynthesised silver nanoparticles have been shown to be antibacterial in nature against a number of dental pathogens such as *S. mutans* and *Lactobacilli* species [12]. Previous studies have reported on the incorporation of chemically synthesised silver nanoparticles into denture acrylic [7, 13]. In order to prevent denture stomatitis and treat recurrent infections, the incorporation of antimicrobial agents into denture base resin and resilient liners have also been reported [7, 13–15].

## Why is it important to do this systematic review?

There is no gold standard of care in the management of denture stomatitis [15]. It is usually managed by oral hygiene instruction, denture hygiene education, correction of denture wearing habits and even dietary therapy. Topical antifungal therapy such as the use of Nystatin is commonly used in the management of denture stomatitis [15]. However, recurrent infections are common as well as the potential for the development of antifungal resistance. The use of topical antifungals is further complicated by the continuous removal of antifungal action by saliva and swallowing, lack of patient compliance and the persistent contact between the infected denture base and affected tissues [15].

Modification of denture bases and denture liners with silver nanoparticles present a unique opportunity to manage denture stomatitis. It is advantageous as the 'modified' dentures are in contact with the tissue thereby preventing re-infection. Sustained release of a drug from the denture base could further prevent the development of a biofilm and the colonisation by *Candida*. Nanomedicine refers to the use of nanotechnology in the diagnosis of disease, drug design and delivery, and implants [8–10, 16]. The materials commonly used to develop these nanotechnology products are inorganic and metal nanoparticles, carbon nanotubes, liposomes, and metallic surfaces [16]. The effect of silver nanoparticles on the human body, more specifically the oral environment, has not yet been fully explored but this systematic review will assist in identifying and investigating these further.

The biological methods of synthesizing silver nanoparticles, referred to as green synthesis, are also readily available and allows these to be used in biomedical applications, such as in dentures or denture liners [17]. As stated above, biological synthesis of silver nanoparticles has several encouraging prospects from profitability to easy manufacturing and being naturally safe with a single step process for large scale synthesis of nanoparticles [11].

## Aim

The aim of this systematic review is to determine if silver nanoparticles inhibit the growth of *Candida Albicans* when included in acrylic dentures and in different denture liners (temporary or permanent types).

The research question using the PICO (Participants, Intervention, Comparator, Outcomes) format for this SR is: *In adults, does silver nanoparticles inhibit the growth of candida albicans in dentures and denture liners compared to normal treatment options?*

## Objective

The objective for this systematic review will include identifying studies where silver nanoparticles included in either denture base acrylic or in denture liners had an effect on the growth of *Candida albicans*, and thus prohibits the development of denture stomatitis.

## Methods

The preferred reporting system will follow the PRISMA-P (Preferred Reporting Items for Systematic Reviews and Meta-Analysis) guidelines [18]. This protocol was registered with the International Prospective Register of Systematic Reviews (PROSPERO): CRD42019145542 and with the University of the Western Cape (UWC) institutional ethics committee (Registration No: BM20/4/1).

The criteria for considering studies to be included for this SR are identified according to study design, type of participants or samples included, and a detailed description of the intervention and outcomes related to the objectives (Fig 1) [19]. Context and setting includes clinical dental treatment and/ or any laboratory studies where silver nanoparticles were used within denture acrylic resin or denture liner materials.

## Types of studies

*Primary clinical and laboratory research studies*, *for example*, *in vitro* and *in vivo* human studies will be included where these have investigated silver nanoparticle inclusion in denture acrylics or denture liners as used in Prosthodontics.

## Types of participants/samples

Studies that report on male and/ or female adult human participants and *in vitro* research will be included. For in vitro studies, preparation of samples of materials will be reported.

## Types of interventions

### Intervention

Complete dentures or denture liner materials incorporated with silver nanoparticles that were synthesised using different methods.

### Control

Complete denture acrylics or different denture liners with no modifications, that is, no inclusion of any AgNPs within these materials.

### Types of outcome measures

The outcomes pre-specified for this SR focuses on the following areas of interest as it relates to the objectives:

### Primary outcomes

Resolution of denture stomatitis with the inclusion of silver NPs in the acrylic denture base or denture liners (temporary or permanent). If an existing infection was present the silver NPs a new denture acrylic denture base or denture liner could allow for the tissues to rehabilitate.

## --- STUDY ELIGIBILITY FORM ---

Reviewer ID:                           Date Reviewed:

Reference/ Study ID: ___________________________________________

_________________________________________________________________

|  | Yes | Unclear | No |
|---|---|---|---|

a)  *Type of Study*:

In vivo; In vitro; animal studies

**Exclude Study type**

b)  *Trial Intervention/ Control:*

Silver Nanoparticles

**Exclude**

c)  *Trial Participants:*

Humans; Animals; Material samples

**Exclude**

d)  *Any other Reasons for Exclusion:*

Not Related to Prosthodontic Treatment

of Denture Stomatitis for denture / reline patients

**Include (**Subject to clarification)                 **Exclude**

e)  *Final Decision:*

**Include**          **Unclear**          **Exclude**

**Fig 1. Nanoparticle systematic review study eligibility form.**

### Secondary outcomes

The efficacy of silver NP on fungal activity by reporting a reduction in colony forming units (CFUs); there are many methods to determine the antimicrobial effect of the modified biomaterial and this is only but the simplest technique.

Effect of silver NP on the prevention of biofilm formation on the denture acrylic or denture liner; prevention of biofilm formation would assist patients who are at risk and struggle with appropriate denture hygiene methods.

Effect of silver NP on *Candida albicans;* the exact mechanism of action is not known but it does result in loss of viable cells.

## Search methods for identification of studies

### Electronic searches

A computerized search will be conducted for primary and ongoing studies to identify literature on the topic of silver NP included in acrylic denture materials or denture liners as a treatment strategy against candida infections as applied in prosthodontics. The databases that will be accessed include: EbscoHost, PubMed, Wiley, Scopus, Sciencedirect and only studies reported in English for the period 2000 to 2020 will be included. Key terms will be combined using Boolean operators and MESH terms for search strategies which will be specific for each database and will be developed using the database specific functions [20]. Medical subject headings will be applied in databases that will allow this function [20]. The search will be adapted to each database to eliminate any inconsistencies that may affect data extraction. Following searches, duplication of results will be eliminated using Mendeley and the 3-step identification of studies (titles, abstracts and full texts) will be completed as indicated on the PRISMA-Flow chart [18]. These references will be managed using Rayyan QCRI, a web tool designed to assist researchers conducting systematic reviews.

A broad search strategy will be used and it will focus on how AgNps were incorporated within the denture materials and the different primary research study designs to determine the effect of AgNps on *Candida albicans. One example of such a search strategy for dentures includes*: (denture* OR denture acrylic OR complete dentures OR complete acrylic dentures) AND (silver nanoparticles) AND (*Candida albicans*) AND (clinical trials, *in vitro* and *in vivo* studies, longitudinal studies OR observational OR randomized controlled trials) AND (2000–2020) [20].

### An example of a search strategy for denture reline materials includes

(denture liners OR resilient liners OR denture relines OR tissue conditioners) AND (silver nanoparticles) AND (*Candida albicans*) AND (clinical trials, *in vitro* and *in vivo* studies, longitudinal studies OR observational OR randomized controlled trials) AND (2000–2020) [20].

Further hand-searching will be conducted including citations from reference lists of retrieved studies for additional references. Where full texts are unavailable, authors will be contacted for unpublished reports of conference proceedings or missing data, where needed. Where registries are available for on-going studies, these will also be considered. In addition, experts in the field of research related to the silver NP will also be contacted. Moreover, the UWC school of Dentistry, which has completed research on silver NP, will be contacted for any recent information related to the topic. The results of searches will be reported using the PRISMA Flow chart [18].

## Data collection and analysis

### Selection of studies

Two review authors (SK and RA) will independently search and then screen the results of the searches which will be recorded on Rayyan QCRI. They will independently complete the study eligibility and data extraction forms for each included study (Figs 1 and 2) [19]. Where eligibility is unclear, clarification will be sought and the differences will be resolved by consultation between the review authors. All types of reviews or synthesis research will be excluded. Primary studies that did not meet the inclusion criteria will be excluded and the reasons for exclusion will be reported.

### Data extraction and management

Two review authors (SK and RA) will independently extract information on study methods, participants, interventions, outcomes, and conclusions from each included primary study using a specially designed pre-piloted data extraction form (Fig 2) [19]. The risk of bias to assess the methodological quality of each included clinical study will independently be completed by the 2 reviewer authors (SK and RA), where possible [19]. The authors will also critically appraise the included studies using a specially designed appraisal tool for laboratory studies. The criteria for this tool follow the guidelines of two similar tools [21, 22]. These criteria are included in Table 1, and scoring system followed similar measures as the articles referenced [21, 22].

### Quality assessment of included studies

The reliability of results of studies depends on the extent to which potential sources of bias has been reduced or avoided when planning the study. The inclusion criteria for this SR will include studies of different designs, thus the quality assessments tools used should be appropriate for these studies. For this SR protocol, a tool for clinical research and one for laboratory studies are described in detail.

**A: Risk of Bias tool for clinical studies [19].** Risk of bias (RoB) is normally used to assess clinical trials. We included a risk of bias assessment of included clinical studies on silver nanoparticles for this SR (Table 1) [19]. The assessment will be completed across the following six components of the RoB tool:

1. Randomization: Sequence Generation:
   For this aspect, the types of randomization sequence used in the studies will be recorded, whether it was block type and/or whether stratification was included or not.

2. Randomization: Allocation Concealment
   Different ways of allocation of interventions to address prevention of foreknowledge must be included, using independent research assistants to complete such as concealed envelopes and centrally conducted allocations may be employed for clinical studies to reduce or eliminate selection bias.

3. Blinding:
   It is also referred to as masking and here different people may be blinded, such as the study participants, researcher and data analyst, this is purely dependent on the nature of the treatment. Including this aspect allows avoiding any negative effects on the study outcomes and outcome measurement.

4. Incomplete Outcomes Data
   Incomplete reporting raises suspicion that the effects are biased. Thus, inclusion of an 'intention-to-treat' principle in the analysis assists with attrition bias.

Ethnicity: ____________________________________________________________________

____________________________________________________________________________

### D. INTERVENTIONS (I) and OUTCOMES

|                    | Total I | Specific I | Duration | Details |
|--------------------|---------|------------|----------|---------|
| **Experimental Group** |     |            |          |         |
| **Control Group**  |         |            |          |         |
| **Laboratory Group** |       |            |          |         |
|                    |         |            |          |         |

Primary Outcomes: _________________________________________________________

____________________________________________________________________________

Secondary Outcomes: _______________________________________________________

____________________________________________________________________________

Adverse Events: ___________________________________________________________

### E. RESULTS

|                        | N | Missing | Summary Data | Measure Effect | Subgroup Analysis |
|------------------------|---|---------|--------------|----------------|-------------------|
| **Experimental Group:** |  |         |              |                |                   |
| **Control Group:**     |   |         |              |                |                   |
| **Laboratory Group**   |   |         |              |                |                   |
| **Outcomes:**          |   |         |              |                |                   |
|                        |   |         |              |                |                   |
|                        |   |         |              |                |                   |

### F. NOTES

Conclusions: ______________________________________________________________

Funding: __________________________________________________________________

Correspondence Needed: ____________________________________________________

____________________________________________________________________________

____________________________________________________________________________

Conflict of interest Statement:

____________________________________________________________________________

**Fig 2. Nanoparticle systematic review data extraction form.**

5. Selective Reporting
   Researchers tend to report only significant results and this type of publication bias is what is referred to here.

6. Other Bias
   There may be sources of bias found under certain specific circumstances which may be reported, such as in cross-over trials.

Each of the 6 criteria is judged as 'Yes', 'No' or 'Unclear' and these depend on the reporting of these within the studies. These correspond to a study obtaining a score or description of having a 'low" 'high' or 'unclear' risk of bias respectively, where having a high risk of bias implies that the criteria were not met, making it research of poor quality. Where the information in the article is insufficient for making judgements based on the RoB tool, authors may be contacted for clarification.

**B: Quality assessment of laboratory studies [21, 22].** The second critical appraisal tool that will be used is for the laboratory studies; it was created by the authors using two others sources as a guide (Table 2) [21, 22]. The parameters to be used are (1) standardization of sampling procedures, (2) description of sample size calculation, (3) calibration of sample before applying the test/ test design in accordance with standards and specifications such as International Standard Organization (ISO) [23], (4) evaluation of results and (5) the use of appropriate statistical analysis.

With regards to the scoring system the criteria for obtaining a particular score is explained as follows: For each of the 5 criteria, if the paper reports it or not, the scores must be recorded following these guidelines and these are then given the appropriate score. A score = 0 is given if the article clearly reported the specific parameter; A score = 1, is recorded for a particular parameter if it was reported, but insufficient details were provided or it was reported unclearly; and A score = 2, is recorded if it was not possible to find the information related to a specific parameter [21, 22].

Following this, a total score will be given and the interpretation of these scores are as follows: A score of 0 to 3: *Low Risk of Bias* for a study is implied when it obtains such a score; A score of 4 to 7: *Moderate Risk of Bias* label is given to a study where this score was recorded; and A score of 8 to 10: *High Risk of Bias* for a study where most criteria were not met, making it a study of poor quality [21, 22].

**Table 1. Risk of bias tool to assess methodological quality of included clinical studies.**

| Features of Risk of Bias | SCORING SYSTEM | | |
| --- | --- | --- | --- |
| | Yes | No | Unsure |
| **Randomization**: Sequence Generation (Selection Bias) | Sequence of randomization recorded | Not completed | Unclear records |
| **Randomization**: Allocation Concealment (Selection Bias) | Record of how the treatment was allocated | | |
| **Blinding** (Detection and Performance Bias) | Who was included is recorded | | |
| **Incomplete Outcomes Assessment** (Attrition Bias) | Outcomes clearly recorded | | |
| **Free of Selective Reporting** (Reporting Bias) | All aspects considered bias addressed | | |
| **Free of other sources of Bias** | Clearly recorded | | |
| INTERPRETATION OF THE SCORING SYSTEM | **LOW RISK OF BIAS** | **HIGH RISK OF BIAS** | **LACK OF INFORMATION** |

**Table 2. Critical appraisal tool to assess methodological quality of included laboratory studies.**

| | QUALITY ASSESSMENT TOOL FOR SILVER NANOPARTICLE LABORATORY STUDIES | | | | |
|---|---|---|---|---|---|
| | | | SCORES | | TOTAL SCORES |
| | Assessment Criteria | 0 | 1 | 2 | |
| 1 | Standardization of obtaining of samples | | | | |
| 2 | Description of sample size calculation | | | | |
| 3 | Performance of specimen dimensioning, test designs, according to standard specifications such as ISO or ASTM | | | | |
| 4 | Were the outcomes measured in a valid and reliable way/testing types according to standard specifications such as ISO or ASTM? | | | | |
| 5 | Was appropriate statistical analysis used? | | | | |

KEY: ISO: International Standard Organization.

ASTM: American Society for Testing Materials.

The extracted data from each study using the form as a guide will be recorded on a Table of Characteristics of included studies (Fig 2). Information retrieved from the included studies will also include authors, titles, country, setting, source of publication, number of patients or specimens studied, where appropriate and other particular characteristics such as final specimen size, study methods utilised, and statistical analyses according to the inclusion and exclusion criteria. In addition, funding sources of projects, ethical clearance, conclusions, comments and correspondence required will also be extracted. Study authors will be contacted in the case of unclear or missing data and any disagreements will be resolved by consensus between review authors.

## Preferred Reporting Items for Systematic Reviews and Meta-Analyses (PRISMA) checklist [18]

SRs and meta-analyses are essential when summarizing evidence related to the efficacy of materials, clinical procedures and interventions reliably and accurately. Therefore, the transparency of conducting this type of synthesis research is important but it unfortunately has not been reported optimally across disciplines. Poor reporting of SRs diminishes their value to clinicians, policymakers and patient treatment outcomes. For this review, the PRISMA P checklist will be completed. Similarly, once the SR has been completed, it is advisable to report the steps using checklists such as the PRISMA statement to indicate that all stages were strictly followed as set out for this study and in this protocol [18].

## Data synthesis and analysis

The results, after critical appraisal of all the included studies will be collated, synthesised and reported in a meta-analysis, if possible. Subgroup analysis will also be considered if appropriate. If not, it will be reported in the form of a narrative. The major entity being analysed (unit of analysis) in this SR will be acrylic dentures and the different denture liners [24].

## Dealing with missing data

For missing data, authors of studies will be contacted to obtain relevant missing information. Analyses will be completed according to the intention-to-treat (ITT) principle where there is no missing data [19, 24]. In the case of missing data, available case analyses will be carried out.

### Assessment of reporting biases

Evidence of publication bias will be reported by constructing funnel plot, if possible.

## Discussion

Traditionally treatment decisions in dentistry were based on empirical knowledge and anecdotal evidence, in addition to in-vitro studies of dental materials. Recently, the importance and value of the research designs such as randomized clinical trials and synthesis research, for example, systematic reviews were gaining more exposure [19]. The emphasis thus changed to conducting quality research, such as systematic reviews, where the evidence may be translated into clinical practice; serving as a guide to practitioners to improve treatment and care to patients. With regards to materials research, many researchers start at the laboratory investigations and very few subsequently get involved with related clinical research. The reasons for this approach are probably related to how feasible the outcomes may be in clinical studies; long observation periods in clinical research; ethical requirements with patient-related research; exorbitant costs associated with clinical research and difficulties with implementing rigorous design principles.

The value of laboratory investigations is not all negative, and by following a strict protocol for re-evaluating these studies, the quality may be determined for such an important subject matter as the use of nanoparticles for dental treatment. That is the purpose of a systematic review. Due to its rigorous design and stringent requirements such as search criteria and critical appraisal, the importance of nanoparticles use in dentistry may be highlighted.

The contribution from this systematic review, knowing that mostly laboratory investigations were completed, may guide investigators (laboratory or clinical) to either include or exclude the use of nanoparticles in dentistry depending on the quality of the included studies and their outcomes. The subject matter may add great value to daily clinical practice if it is determined that nanoparticles may influence the oral effects of *candida albicans*.

By publishing the protocol, it is an indication of being guided using strict ethical principles and can serve as a guide to a rigorous study as in this case the systematic review.

## Limitations of SR

The potential limitations of the SR will focus on several aspects of the study design, searches, critical appraisal and the subject matter. Depending on the type of study designs of included studies, the quality of the extracted information obtained will be important with regards to its use in practice, which refers to the translation of completed research into clinical practice. Only high quality clinical studies are considered translatable into clinical practice. The quality of the research may also have an impact on the teachings if it is or is not of a high standard. This aspect therefore makes reference to the clinical versus laboratory research which will impact on the SR quality and thus on its translation into practice. The other limitation may be the searches as databases have their own constraints. In addition, the critical appraisal tool is a combination of two similar instruments and the challenge will be observed only once it has been used as all areas may be not be answered for all studies. Thus, a final score may not be a conclusive indication of the quality of the SR. In addition, the subject matter is fairly novel in dentistry, thus the areas of research may not address all the clinical research as expected from such a SR.

## Concluding remarks

With the set rigorous inclusion criteria and searching strategies, appropriate clinical and laboratory studies may be obtained when searching databases. But, the analysis, its interpretation

and synthesis will depend on the quality of the research data. For this protocol, appropriate critical appraisal tools were obtained which will assist with this interpretation and conclusions.

## Supporting information

**S1 Checklist. PRISMA-P 2015 checklist.**
(PDF)

**S1 Fig. PROSPERO registration.**
(PNG)

**S1 File. PROSPERO.**
(PDF)

## Author Contributions

**Conceptualization:** Razia Z. Adam, Saadika B. Khan.

**Validation:** Razia Z. Adam, Saadika B. Khan.

**Visualization:** Razia Z. Adam, Saadika B. Khan.

**Writing – original draft:** Razia Z. Adam, Saadika B. Khan.

**Writing – review & editing:** Razia Z. Adam, Saadika B. Khan.

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
