## [Decision Letter · Decision Letter 0]

30 Nov 2020

PONE-D-20-25599

Antimicrobial efficacy of silver nanoparticles against Candida infections in dentures: Systematic Review Protocol

PLOS ONE

Dear Dr. Khan,

Thank you for submitting your manuscript to PLOS ONE. After careful consideration, we feel that it has merit but does not fully meet PLOS ONE’s publication criteria as it currently stands. Therefore, we invite you to submit a revised version of the manuscript that addresses the points raised during the review process.

When submitting the revised version, please, attach the record of the PROSPERO registration of this systematic review protocol as a supplementary file.

We look forward to receiving your revised manuscript.

Kind regards,

Ahmed Negida, MD

Academic Editor

PLOS ONE

Journal Requirements:

2. Please amend either the title on the online submission form (via Edit Submission) or the title in the manuscript so that they are identical.

4. Please include your tables as part of your main manuscript and remove the individual files. Please note that supplementary tables should be uploaded as separate "supporting information" files.

5. Please include captions for your Supporting Information files at the end of your manuscript, and update any in-text citations to match accordingly. Please see our Supporting Information guidelines for more information: http://journals.plos.org/plosone/s/supporting-information

Reviewers' comments:

Reviewer's Responses to Questions

**Comments to the Author**

1. Does the manuscript provide a valid rationale for the proposed study, with clearly identified and justified research questions?

Reviewer #1: Partly

Reviewer #2: Yes

Reviewer #3: Yes

2. Is the protocol technically sound and planned in a manner that will lead to a meaningful outcome and allow testing the stated hypotheses?

Reviewer #1: Yes

Reviewer #2: Partly

Reviewer #3: Yes

3. Is the methodology feasible and described in sufficient detail to allow the work to be replicable?

Reviewer #1: Yes

Reviewer #2: No

Reviewer #3: Yes

4. Have the authors described where all data underlying the findings will be made available when the study is complete?

Reviewer #1: Yes

Reviewer #2: No

Reviewer #3: No

5. Is the manuscript presented in an intelligible fashion and written in standard English?

Reviewer #1: Yes

Reviewer #2: Yes

Reviewer #3: Yes

6. Review Comments to the Author

You may also provide optional suggestions and comments to authors that they might find helpful in planning their study.

Reviewer #1: Review comments

Background

1st paragraph, 5th line, in 2013, it was reported that … comment where was it reported (in Africa, the US etc or where)

Line 9… lack of access or limited access to dental care is strongly associated with edentulism (provide a source to factual statements)

2nd paragraph line 4 the imbalance of oral microbes in the oral cavity is then…source?

3rd paragraph line 2nd… Candida Albicans is the most common causative pathogen of oral…source?

7th paragraph (Why is it important to do this systematic review?), 1st line…There is no standard of care in the management of denture stomatitis…source?

8th paragraph, 5th line: Nanomedicine refers to the use of nanotechnology in the diagnosis of disease...source?

The gap statement on page 5, 3rd paragraph, line 7 to 9 does not match with the aim of the study. 1. will this systematic review harmonize studies by looking at the effect of nanotechnologies on the human body or it will only look at silver nanoparticles on the human body?

2. kindly draft the gap statement such that, it is clear to understand the knowledge gap and what to expect in this study.

If the systematic review aims to determine if silver nanoparticles inhibit the growth of candida albicans, then there should be a literature/ knowledge gap established and it is for this reason you embark on the systematic review to extend knowledge in this area.

Objective

Have a SMART objective free from abbreviations. Kindly write SR and AgNPs in full

Methods

Page 9, 3rd paragraph, 5th line …first write in full the name of the University (UWC) before you can later abbreviate

Grammar

Efficiency should be the efficiency, introduce an article “the”

Early loss should be the early loss

“is” should be changed to “are” Denture hygiene, appropriate mouth rinses and the use of …. “are” instead of “is”

Nontoxic should be hyphenated

The use of these chemicals may limit… you are referring to several chemicals hence the statement should be changed to “their” instead of “its”

The biological methods are cost-effective, environmental friendly and includes….change “includes” to “include”.

Page 4, 2nd paragraph, line 6; the synthesis of silver nanoparticles…replace “have been shown to be to “are”

Page 5, 1st paragraph 1st line introduce an article “the” before the word infected denture

page 11, point number 6 (other bias) line 3…each of the 6 criteria are…replace “are” with “is”…NB: kindly, pay attention to subject-verb agreement

point number 6 (other bias) line 6…criteria was…replace “was” with “were”

page 12 last paragraph, 1st line, “as guide” should be changed to “as a guide”

Reviewer #2: The manuscript aims at determining if Silver nanoparticles inhibit the growth of Candida Albicans when included in acrylic dentures and in different denture liners. However, the following are my comments:

Major Comments:

The abstract needs to be rewritten to be able to stand alone. It should include brief introduction, methods and analysis, and expected results.

In the background authors identified that imbalance of oral microbes in the oral cavity is affected by advanced age. Hence, the age of the participants must be clearly stated. Also, why is the inclusion of animal studies here?

The PRISMA-P must be strictly followed to drive home the points the protocol set to deliver.

For search strategy, I think the draft must include at least one well defined electronic database e.g. PubMed that will be replicated for other databases.

No concluding remarks. The authors should include conclusion to summarise the findings of the protocol.

Minor Comments

The manuscript needs to be properly proofread

The background should be more detailed

Reviewer #3: The authors wrote an excellent protocol for the systematic review on the use AgNPs technology in denture and denture liners and its efficacy against Candida albicans.

A few suggestions for improvement and some questions I have are listed below

Suggestions and questions

1. While the author made their case in a sensible manner as to the significance of conducting this systematic review, It is essential that the claims made on the back ground and significance section be referenced with appropriate literature

E.g.1“The lack of access or limited access to dental care is strongly associated with edentulism. In low-income countries, early loss of teeth due to extractions are common. Extraction of the offending tooth is often the only acceptable, affordable and available option to treat carious teeth for these communities”.

E.g.2 “Topical antifungal therapy such as the use of Nystatin is commonly used in the management of denture stomatitis. However, recurrent infections are common as well as the potential for the development of antifungal resistance. The use of topical antifungals is further complicated by the continuous removal of antifungal action by saliva and swallowing, lack of patient compliance and the persistent contact between infected denture base and affected tissues.”

While introducing the research question with a PICO method, the authors used the term Adult in the beginning of the question. Is this in reference to human trials? If so it will be helpful for the reader if it is also reflected in the sample/participants section.

2. On electronic search, The author expressed that search strategies will be tailored according specific data bases, is author referring that this will also insure de-duplication or has the author plans to undergo a deduplication process after search? I suggest the author clarify this in the text.

3. While the authors presented a compelling discussion on the significance of conducting the systematic review and what it can contribute on this particular area of science in the discussion section, There were no discussion of the potential challenges/limitations. It will be help full for the readers to discuss if there are any potential limitations to conduct the systematic review. How does the authors plan to pull findings from human studies and laboratory studies? Does the variation in the study types pause a challenge while pulling the findings?

7. PLOS authors have the option to publish the peer review history of their article (what does this mean?). If published, this will include your full peer review and any attached files.

Reviewer #1: **Yes: **Edward Agyemang

Reviewer #2: No

Reviewer #3: **Yes: **Yohannes T, Amera

---

## [Author Response · Author response to Decision Letter 0]

9 Dec 2020

I have completed the comments from all Reviewers (1 to 3) and the Editor 

Please see the Rebuttal letter for the detailed comments

---

## [Decision Letter · Decision Letter 1]

8 Jan 2021

Antimicrobial efficacy of silver nanoparticles against Candida  Albicans: A Systematic Review Protocol

PONE-D-20-25599R1

Dear Dr. Khan,

We’re pleased to inform you that your manuscript has been judged scientifically suitable for publication and will be formally accepted for publication once it meets all outstanding technical requirements.

Kind regards,

Ahmed Negida, MD

Academic Editor

PLOS ONE

Additional Editor Comments (optional):

Reviewers' comments:

Reviewer's Responses to Questions

**Comments to the Author**

1. Does the manuscript provide a valid rationale for the proposed study, with clearly identified and justified research questions?

Reviewer #1: Yes

Reviewer #3: Yes

2. Is the protocol technically sound and planned in a manner that will lead to a meaningful outcome and allow testing the stated hypotheses?

Reviewer #1: Yes

Reviewer #3: Yes

3. Is the methodology feasible and described in sufficient detail to allow the work to be replicable?

Reviewer #1: Yes

Reviewer #3: Yes

4. Have the authors described where all data underlying the findings will be made available when the study is complete?

Reviewer #1: Yes

Reviewer #3: Yes

5. Is the manuscript presented in an intelligible fashion and written in standard English?

Reviewer #1: Yes

Reviewer #3: Yes

6. Review Comments to the Author

You may also provide optional suggestions and comments to authors that they might find helpful in planning their study.

Reviewer #1: Initially there were major grammatical mistakes. However, the mistakes are rectified. Authors should follow keenly the methodology spelled out

Reviewer #3: The authors have amended the feed backs I have provided in my previous review and the manuscript has improved immensely.

7. PLOS authors have the option to publish the peer review history of their article (what does this mean?). If published, this will include your full peer review and any attached files.

Reviewer #1: **Yes: **Edward Agyemang

Reviewer #3: **Yes: **Yohannes Tesfaye Amera

---

## [Editor Report · Acceptance letter]

14 Jan 2021

PONE-D-20-25599R1 

Antimicrobial efficacy of silver nanoparticles against Candida Albicans: A Systematic Review Protocol 

Dear Dr. Khan:

I'm pleased to inform you that your manuscript has been deemed suitable for publication in PLOS ONE. Congratulations! Your manuscript is now with our production department. 

Kind regards, 

on behalf of

Dr. Ahmed Negida 

Academic Editor

PLOS ONE